# Unraveling Large and Polyploidy Genome of the Crucifer *Orychophragmus violaceus* in China, a Potential Oil Crop

**DOI:** 10.3390/plants12020374

**Published:** 2023-01-13

**Authors:** Qi Pan, Pan Zeng, Zaiyun Li

**Affiliations:** 1Resource Institute for Chinese & Ethnic Materia Medica, Guizhou University of Traditional Chinese Medicine, Guiyang 550025, China; 2College of Plant Science and Technology, Huazhong Agricultural University, Wuhan 430070, China

**Keywords:** *Orychophragmus*, germplasm characterization, plant wild relatives, polyploidy, oilseeds

## Abstract

The genus *Orychophragmus* in the *Brassicaceae* family includes the types with 2n = 20, 22, 24, and 48. The species *O. violaceus* (L.) O. E. Schulz has 2n = 24 and is widely cultivated as an ornamental plant in China. This review summarizes the research progress of its genome structure and evolution in the context of cytogenetics and genome sequencing. This species has a large genome size of ~1 Gb and longer chromosomes than those of *Brassica* species, which is attributable to the burst of TE insertions. Even more, one tetraploidization event from about 600–800 million years ago is elucidated to occur during its genome evolution, which is consistent with the polyploidy nature of its genome revealed by the meiotic pairing patterns. Its chromosomes are still characterized by a larger size and deeper staining than those from *Brassica* species in their intergeneric hybrids, which is likely related to their inherent differences between genome structures and cytology. Its genome is dissected by the development of additional alien lines, and some traits are located on individual chromosomes. Due to the abundant dihydroxy fatty acids in its seed oil with superior lubricant properties and wide environmental adaptations, this plant promises to be utilized as one new oil crop in the future.

## 1. Introduction

Chinese violet cress (*Orychophragmus violaceus* (L.) O. E. Schulz, also called Zhuge Cai) is an annual or biennial herb of the genus *Orychophragmus* in the *Brassicaceae* (*Cruciferae*) family. Its Chinese name Zhuge Cai (Zhuge vegetable) originated from the legend that the young stems of this plant were used to make up for a food deficiency by Prime Minister Zhuge Liang’s army in a northern expedition during the Three Kingdoms dynasty of ancient China, ca. 220–280. *O. violaceus* is native to China and is widely distributed in different ecological regions, and its wild species is also found in North Korea [1]. As its beautiful purple flowers bloom around the second month of the Chinese lunar calendar, *O. violaceus* is also known as the February orchid (eryuelan) (*Moricandia sonchifoli* (Bunge) Hook. f.). As the plants in the *Orychophragmus* genus display wide variations in morphology, their classifications are still under revision [2,3,4]. Interestingly, the plant types with 2n = 20, 22, and 48 have been collected and classified, except for the popular one with 2n = 24 [2,3,5,6] (Table 1). The most common Zhuge Cai (*O. violaceus* (L.) O. E. Schulz) has 2n = 24. The type with 2n = 48, named *O. taibaiensis,* was found only in Taibai Mountain in Shanxi Province, Northwest China, which might be an autotetraploid originating from the chromosome doubling of the 2n = 24 plant [2,3]. Plants with 2n = 20 and 22 were classified as *O. diffusus* and *O. hupehensis*, respectively, and both have specific morphological characteristics and a smaller plant architecture than that of *O. violaceus*, which results likely from the reduced chromosome numbers. *O. hupehensis* was collected at several locations with different altitudes in Hubei Province, Central China, while *O. diffusus* was only found up to now in the area around Shanghai with low altitude, East China. So the loss of certain chromosomes was assumed not only to be associated with their phenotype change but also with the limited ecological distributions.

Besides the taxonomic identifications, the *Orychophragmus* plants have also been characterized for their cytogenetics, genome structure, and evolution and used as the germplasm for the genetic improvement of *Brassica* crops during the last several decades. Recently, the genome sequencing of *O. violaceus* (2n = 24) was completed by two groups and revealed the specific genome duplication event during the evolution [7,8]. Here we aim to review the significant progresses in these aspects and to give meaningful suggestions for future study.

## 2. Phylogenetics

In the previous molecular phylogenetic tree, *O. violaceus* seemed to form a unique clade and was called the floating genera together with some other crucifers [9]. Based on the genome comparisons among *O. violaceus* and 12 other sequenced *Brassicaceae* species and the phylogenetic tree constructed, *O. violaceus* was found to be close to *Brassica* and *Schrenkiella parvula* in *Brassicaceae* lineage II but distant from *Arabidopsis* (lineage I) [7,8] (Figure 1). However, no genomic evidence supported that it was descended from the tetraploid intermediate *Brassica* ancestor, indicating their independent evolutionary paths [8].

## 3. Large and Polyploidy Genome

*O. violaceus* (2n = 24) has a genome size of ~1.3 Gb, which is larger than those of many other cruciferous species [7,8], even than three *Brassica* allotetraploid species, *Brassica juncea* (genomes AABB, 922 Mb), *B. napus* (AACC, 1200 Mb), and *B*. *carinata* (BBCC, 1150 Mb) [10,11,12], than the double of three *Brassica* diploids, *B. rapa* (AA, 529 Mb), *B. nigra* (BB, 570 Mb), *B. oleracea* (CC, 630 Mb) [13,14,15]. The large genome was mainly caused by the burst of transposable elements (TEs) insertions, and about one-half of the genome sequences were annotated as LTR-RTs. The lineage-specific burst of LTR-RT insertions, which occurred ~0.57 million years ago (MYA), was estimated to generate ~540 Mb LTR-RTs, contributing substantially to its larger genome [8]. Its chromosomes have similar lengths, most have metacentric centromeres, and a few have submetacentric centromeres [16,17] (Figure 2a). In mitotic metaphase, with the traditional dye Fuchsin, the chromosomes are evenly stained, except for their centromeres and secondary constrictions. By genomic in situ hybridization (GISH) analysis using its own genomic DNA as a probe, all chromosomes show the even distribution of signals along the whole length (Figure 2a), indicating that the chromosomes are full of moderately repetitive DNA sequences distributed evenly throughout the entire chromosomes [17]. In contrast, the chromosomes of *Brassica* species show the GISH signals mainly in the centromeric regions. 

The high numbers of both 5S and 45S ribosomal RNA (rRNA) genes might suggest the polyploid origin of its chromosome complement (Figure 2b) [18]. Twenty-two 5S rDNA sites with variable sizes are located at the terminal parts of 18 (9 pairs) chromosomes, of which 4 chromosomes contain double sites of 5S rDNA in their terminal parts—two sites of different fluorescent intensities on one chromosome arm separated by a short distance, with the distal one being stronger than the proximal one. No chromosomes with double 5S rDNA sites carry the 45S rDNA site. Eight 45S rDNA sites of different sizes are detected at the terminal parts of 8 chromosomes, and all eight 45S rDNA sites are co-localized with 5S rDNA sites, with the latter ones being more proximal. As *O. violaceus* has three pairs of satellite chromosomes, the 45S rDNA loci on one of the chromosome pairs have no transcriptional activity. Correspondingly, in the allopentaploid (2n = 50, AACCO) between *B. napus* (2n = 38, AACC) and *O. violaceus,* which contains the haploid complement of 12 *O. violaceus* chromosomes, 3 chromosomes show satellites [19,20] (Figure 2c). All terminal localizations of both rDNA loci show that their positions undergo little changes, in contrast to different positions in *Brassica* species.

The meiotic pairing in pollen mother cells (PMCs) of *O. violaceus* is not fully diploid-like, for the flexible pairing patterns appear, by the formation of the variable multivalents with four, six, or more chromosomes plus the respective number of bivalents, though the pairing of 12 bivalents is prevalent [21,22]. The non-diploidization of its meiotic behavior suggests homoeology among the chromosomes. However, equal chromosome segregation is performed. In consistence, various chromosomal pairings also occur in the PMCs of its haploids, and all chromosomes remain unpaired only at a low rate [23]. Extremely, all chromosomes are paired as six bivalents, or as four trivalents, respectively, which further demonstrates the high degree of homology and polyploidy of the genome, possibly with the basic chromosome number x = 6 or x = 3.

The tetraploidy nature of the genome was early suggested by the finding that the meiotic pachytene chromosomes had two copies of the conserved segments F and U of the ancestral karyotype of the *Brassicaceae* family [24]. From the recent genome sequencing, the genome of its ancestral diploid was revealed to experience one tetraploidization event about 600–800 million years ago, and the event was independent of the hexaploidization event of the *Brassica* diploids [7,8] (Figure 1). The tetraploidization event was close but occurred later than the hexaploidization event in *Brassica* [8]. From the sequence comparisons with other sequenced species, the diploid ancestor of *O. violaceus* most likely kept the tPCK (translocated proto-*Calepineae* karyotype) with n = 7, and the duplicated tPCK with n = 14 derived the extant genome with n = 12, following the chromosomal translocations and inactivation of two centromeres. Comparisons between the two reconstructed subgenomes revealed the subgenome dominance, indicating that two subgenomes likely originated via allotetraploidization events from a wide cross [8] (Figure 1). So expectedly, the homoeologous relationships existed among its chromosomes, and the structural divergence is not enough to prevent the formation of multivalents, or the genetic system which inhibited the homoeologous pairing has not evolved yet.

## 4. Maintenance of Chromosome Features in Wide Hybrids

Due to its larger genome, the larger chromosomes of *O. violaceus* with even staining make them distinguishable from the smaller ones from *Brassica* species in their hybrid cells, even only by the traditional cytological method. In the somatic intergeneric hybrids between *B. napus* and *O. violaceus* (2n = 62, AACCOO) and progenies, the chromosomes of *O. violaceus* were significantly longer than those of *Brassica* species in the same cells [19,20] (Figure 2c,d). From the intergeneric sexual crosses between *O. violaceus* as a male parent and six cultivated *Brassica* species as female parents, hybrids were produced, but only the hybrids with *B. oleracea* (2n = 18, CC) were classical hybrids that had the expected number of parental chromosomes (2n = 21, CO), and the other five were all nonclassical hybrids with unexpected chromosomal complements (Figure 3). The reciprocal crosses with *O. violaceus* as a female parent failed to give hybrids [22,25,26,27,28,29]. These nonclassical hybrids contained cells with variable chromosome numbers and complements: most of the cells had all the chromosomes from *Brassica* parents without or with some from *O. violaceus*, and the remaining had the partial chromosomes of *Brassica* parents without or with some from *O. violaceus* because the chromosomes from *O. violaceus* were largely lost. The common feature in these nonclassical hybrids was the easy distinction of some darkly stained and larger chromosomes in mitotic and meiotic cells, which were assumed to be the origin of *O. violaceus*. The reason for the more apparent manifestation of the difference in chromosome size and staining during meiosis than mitosis was likely that the structural differences between the parental chromosomes could be displayed more obviously during the long duration of the meiotic process [25,26,29].

There are certain differences in the cytological behaviors among these nonclassical hybrids, which depend on the genome types of *Brassica* parents [29] (Figure 3). The different chromosomal behaviors in the hybrids between *O. violaceus* and three *Brassica* diploids might be related to their inherent differences in cytology and genome structure. *B. oleracea* has a genome with a larger size and a higher percentage of repetitive sequences (~60%) than that of *B. nigra* (~50%) and *B. rapa* (30%) [13,14,15]. The ~30% larger genome size of *B. oleracea* than *B. rapa* is attributable mainly to the continuous TE (Transposable Elements) amplifications during the last four million years since their split [15]. All chromosomes from three diploids during prometaphase in mitosis showed clear heterochromatin segments in the centromere regions and lightly condensed regions at the terminal part of some chromosome arms [30]. At the diakinesis stage of meiosis, the bivalents of *B. rapa* exhibited highly condensed and deeply stained centromeric regions, less condensed and lightly stained terminal parts, and very clear metacentric and submetacentric centromeres, which could be used to discriminate each bivalent, especially those with satellites [31]. The bivalents of *B. nigra* also showed highly condensed and deeply stained centromeric regions and terminal parts [32]. In contrast, the bivalents of *B. oleracea* showed no or slight differentiation of chromatin condensation [31]. Such chromosome morphology of *B. rapa* and *B. oleracea* were retained in their allotetraploid *B. napus* and *B. rapa–B. oleracea* alien addition lines with additional chromosomes from *B. oleracea*, which would be the cytological markers for the distinction of the chromosome origin [31,33]. Similar to *B. oleracea*, *O*. *violaceus* showed evenly condensed and stained bivalents [21,22]. Perhaps, it was the similar chromosomal characteristics of *B. oleracea* and *O. violaceus* that ensured the coordination of their chromosomes during cell division in their hybrids and no loss of the *O. violaceus* chromosomes [26,27]. Inversely, the loss of all or partial chromosomes from *O. violaceus* in the hybrids with *B. rapa* and *B. nigra* might be due to their differential chromosomal performance leading to the asynchronous behavior of the parental chromosomes [27]. In addition, the larger genome size of *O. violaceus*, almost double of the *Brassica* diploids, may be another reason for the loss of its chromosomes. Due to the interrelationships among six Brassica species, the cytological behaviors of the hybrids between *O. violaceus* and three *Brassica* allotetraploids seemed to be explainable from those of the hybrids with three diploids (Figure 3). The hybrids with *B. napus* (AACC) and *B. carinata* (BBCC), which shared the C subgenome of *B. oleracea,* had less difference in chromosome size and staining, while the hybrids with *B. juncea* (AABB) showed such difference mostly [22,25,26,29]. The unstable cytology in these hybrids also indicated the distant phylogenetical relationships between the two genera.

## 5. Genome Dissection via Development of Alien Additional Lines

Although the sexual hybrids between *B. napus* and *O. violaceus* were cytologically unstable and lost most of the chromosomes from *O. violaceus*, their somatic hybrids produced by the protoplast fusion contained all the chromosomes of two parents, with 2n = 62 (AACCOO). Then the hybrid was backcrossed by *B. napus* as a recurrent parent, and 9 of the potential 12 alien addition lines were identified from the phenotype and cytology in the progeny populations [19,20] (Table 2). The monosomic alien addition lines (MAALs)/disomic alien addition lines (DAALs) with all 38 chromosomes of *B. napus* and one or two copies of each chromosome from *O. violaceus* were denoted by the codes MAAL1–MAAL9, DAAL1–DAAL9, and expressed some morphological characters of *O. violaceus* or some novel traits unobserved in two parents [19]. MAAL1 had the trait of serrated leaves from *O. violaceus*. MAAL2 was morphologically similar to *B. napus* but had small oval leaves and small flowers and also weaker growth. MAAL3 had flower petals with white and shrunken margins. MAAL4 had yellow flowers, while DAAL4 produced red petals of *O. violaceus* origin. Furthermore, the *OvPAP2* gene was cloned from the additional line and transformed into *B. napus*, resulting in transgenic plants with red flowers [34]. MAAL5 showed the trait of basal branching from *O. violaceus*. MAAL6 bloomed earlier than both parents and was significantly shorter than other addition lines but showed no obvious traits of *O. violaceus*. MAAL7 was completely female sterile but fertile for males, and the gynoecium development was studied by transcriptomic analyses [35]. MAAL8 had the leaves curled upward at the seedling stage. MAAL9 had leaves of light color, but this trait was unstable. The traits of *O. violaceus,* including serrated leaves, basal branching, and purple flowers, were also expressed in the hybrids with *Brassica* species [22,25,26] and were dominant.

The chromosomes from *O. violaceus* in the alien addition lines MAAL4–MAAL6 carried 45S rDNA loci, and the satellites formed nearby (Figure 2d), indicating that these 45S rDNA loci were active in the *B. napus* background [19]. Transcriptions of the rRNA gene of *O*. *violaceus* were detected by SSCP (single-strand conformation polymorphism) method) in these three additional lines, and there was no transcription of rRNA genes from *B. napus* in MAAL4, indicating that the rRNA genes of *O. violaceus* were completely dominant over those of *B. napus*. In MAAL5 and MAAL6, the expressions of the rRNA genes of both *B. napus* and *O. violaceus* were detected, indicating that the rRNA genes of *O. violaceus* were partially dominant. So *O. violaceus* showed both nucleolar dominance and phenotypic dominance over *Brassica* species.

## 6. A Potential Oil Crop

The seed oil of *O. violaceus* was early found to have a high content of linoleic acid (~50%) and a low content of erucic acid (~1%), which provided a new genetic resource of low erucic acid for improving the fatty acid compositions of *Brassica* crops [6,36,37]. A few years ago, it was discovered that its seed oil contained two kinds of abundant 24-carbon dihydroxy fatty acids (7, 18-(OH)2-24: 1∆15; 7, 18-(OH)2-24: 1∆15′ 21), accounting for about 40% of the total fatty acids [38]. These dihydroxy fatty acids were found for the first time in the plant kingdom, which conferred the oil with even better high-temperature lubricant properties than that of castor oil. The candidate genes for their biosynthesis were detected by the analyses of the genomic data [7,8]. Its *Fatty Acid Desaturase2* (*FAD2*) and *Fatty Acid Elongase1* (*FAE1*) were found to develop specific enzymatic activities critical for dihydroxy fatty acids biosynthesis [38]. The enzyme encoded by *OvFAD2-2* functions as a fatty acid hydroxylase to generate the terminal hydroxyl group of two acids instead of catalyzing fatty acid desaturation. Additionally, the functional variants encoded by *OvFAE1-1* produced the carboxyl-terminal hydroxyl group of dihydroxy fatty acids through a “discontinuous elongation” process. Based on the genome and the transcriptome data of multistage seed development, it was predicted that *OvDGAT1-1* and *OvDGAT1-2* (DGAT, diacylglycerol acyltransferase) were the candidate genes regulating the storage of dihydroxy fatty acids in its seeds, but further study is needed to confirm this. From the results of genome sequencing, the formation of dihydroxy fatty acids was hypothesized to occur after the genome tetraploidization event [7]. So this plant is highly promising as a new and high-value industrial oilseed crop, more than one ornamental. Moreover, *O. violaceus* has potential medicinal value, for the seed extract has cytoprotective and hepatoprotective effects [39,40]. Its adaptation to a wide range of ecological environments [41] should contribute to its actual utilization. With the aid of genomic data, specific cytology, and strong in vitro regeneration capability of the cells and tissues [18,42,43,44], its genetic research and breeding would be accelerated in the future.

## Figures and Tables

**Figure 1 plants-12-00374-f001:**
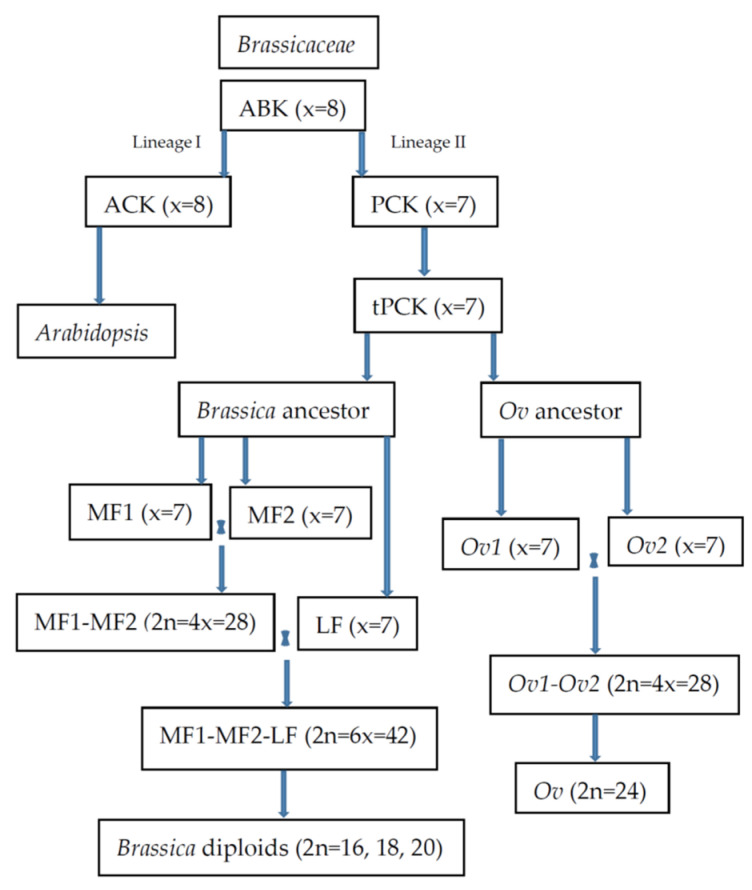
Phylogenetical relationship and karyotype evolution of *Brassica* and *Orychophragmus*. ABK: Ancestral *Brassicaceae* Karyotype. ACK: Ancestral *Camelineae* karyotype. PCK: Proto-*Calepineae* Karyotype. tPCK: translocated Proto-*Calepineae* Karyotype. MF1: Medium Fractionated subgenome. MF2: Most Fractionated subgenome. LF: Least Fractionated subgenome. *Ov*: *O. violaceus*. *Ov1*: *Ov1* subgenome. *Ov2*: *Ov2* subgenome.

**Figure 2 plants-12-00374-f002:**
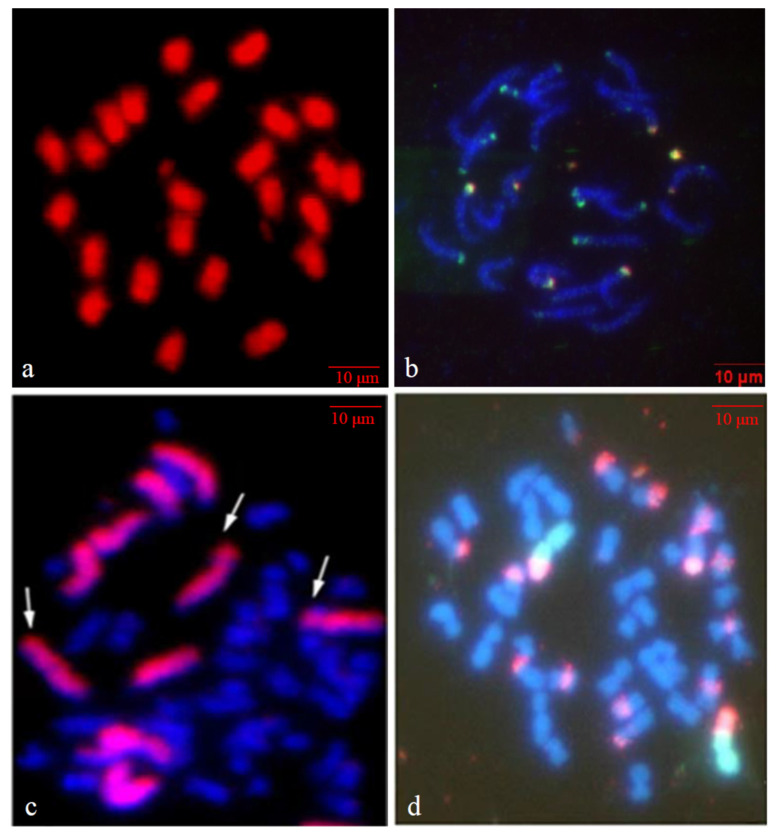
The chromosomes of *O. violaceus* itself and in intergeneric hybrids. (**a**) The chromosomes of *O. violaceus* after GISH with its genomic DNA as the probe, with the homogenous distribution of the red signal from Cy3 along the whole chromosomes. (**b**) The detection of eight 45S rDNA loci (red) and twenty-two 5S rDNA loci (green) by double FISH on the mitotic chromosomes of *O. violaceus* [20]. (**c**) In one prometaphase cell of the pentaploid plant (AACCO) between *B. napus* and *O. violaceus*, the 12 labeled chromosomes from *O. violaceus* are obviously much longer than those from *B. napus*, and the satellites on three chromosomes are obvious (arrows). (**d**) In one metaphase cell of the disomic additional line DAAL4 (2n = 40), the two larger chromosomes from *O. violaceus,* which are labeled green by GISH, carry satellites, but many chromosomes from *B. napus* harbor the rDNA loci (red) but no satellites.

**Figure 3 plants-12-00374-f003:**
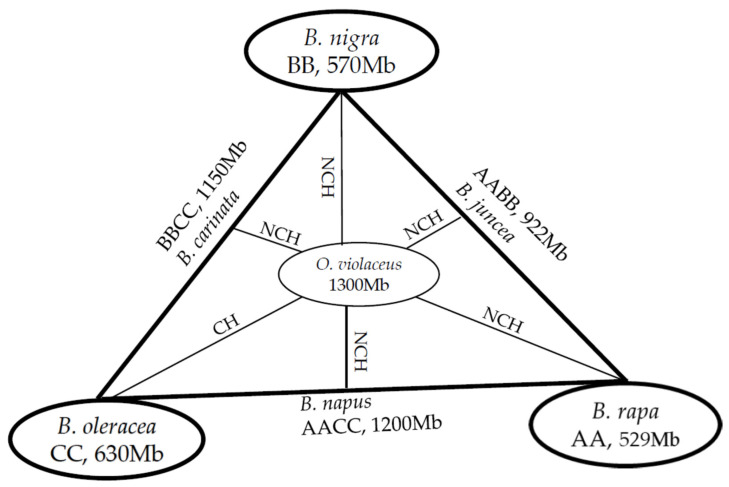
The cytological behaviors of the intergeneric hybrids between *O. violaceus* and six *Brassica* species. The genome size of each species given is the estimated size, not the assembled size. CH: classical hybrids. NCH: non-classical hybrid. The expression dominance of the rRNA genes or the nucleolar dominance from three diploids follows the hierarchy B > A > C in three allotetraploids, with the genes of B. nigra expressed in *B. juncea* and *B. carinata* and the genes of B. rapa in B. napus [29].

**Table 1 plants-12-00374-t001:** The species with different chromosome numbers and their distribution regions in the genus *Orychophragmus*.

Species	2n	Locations	Altitudes (m)
*O. diffusus*	20	Shanghai, East China	50
*O. hupehensis*	22	Hubei Province, Central China	400~1000
*O. violaceus*	24	Over China	Wide ranges
*O. taibaiensis*	48	Taibai Mountain, Northwest China	1200

**Table 2 plants-12-00374-t002:** The phenotype and cytological markers specific to each alien line between *B. napus* and *O. violaceus*.

AALs	Traits
MAAL1/DAAL1	Leaf serration of *O. violaceus.*
MAAL2	Biased to *B. napus* but with small oval leaves and flowers.
MAAL3	Flower petals with white and shrunken margins.
MAAL4/DAAL4	MAAL4 with yellow flowers, DAAL4 with red flowers of *O. violaceus*. Active rDNA loci.
MAAL5	Basal branching of *O. violaceus*. Active rDNA loci.
MAAL6	Early flowering and short plants. Active rDNA loci.
MAAL7	Female sterility.
MAAL8	Leaves curled upward at the seedling stage.
MAAL9	Leaves of light color.

## Data Availability

Not applicable.

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
