# Peer review of "Unraveling Large and Polyploidy Genome of the Crucifer Orychophragmus violaceus in China, a Potential Oil Crop"

_plants, 2023, doi:10.3390/plants12020374_

Round 1
Reviewer 1 Report (Previous Reviewer 1)
The review manuscript submitted by Pan and Li is a re-submission of the previous plant-2039341 and is very much improved. The manuscript describes the genome structure and relationship of Brassicaceae family. The topic described is interesting. I have found some issues as follows that, once addressed, I think that the manuscript will be improved.
It is described that chromosome reduction in O. diffuses and O. hupehensis may be responsible for their specific characteristics in the introduction section. Are there any direct evidences or reports of chromosome reduction in these species? If so, please indicate or cite the results.
Fig. 3. What does the slanted line between B. oleracea and B. rapa mean? If there is some meaningful, it should be explained. It would be easier for the readers to understand if B. juncea, B. carinata and B. napus are enclosed in some form, as well as B. nigra and others.
MAALs are explained in detail, but DAALs are not; a more detailed explanation of DAALs is needed. Personally, I think a table summarizing MAALs would be a better option.
Minor comments
Brassica juncea, B. napus, B. carinata (lines 84-85) must be italic. Same corrections are required in numerous other parts in the manuscript.
Author Response
The review manuscript submitted by Pan and Li is a re-submission of the previous plant-2039341 and is very much improved. The manuscript describes the genome structure and relationship of Brassicaceae family. The topic described is interesting. I have found some issues as follows that, once addressed, I think that the manuscript will be improved.
It is described that chromosome reduction in O. diffuses and O. hupehensis may be responsible for their specific characteristics in the introduction section. Are there any direct evidences or reports of chromosome reduction in these species? If so, please indicate or cite the results.
Response:No such results. The sentence is revised as: So the loss of certain chromosomes was assumed not only to be associated with their phenotype change but also with the limited ecological distributions.
Fig. 3. What does the slanted line between B. oleracea and B. rapa mean? If there is some meaningful, it should be explained. It would be easier for the readers to understand if B. juncea, B. carinata and B. napus are enclosed in some form, as well as B. nigra and others.
Response: The positions of three diploids in the triangle show the hierarchy of expression dominance of their rRNA genes or the nuclealar dominance in the three allotetraploids: B>A>C [29].
MAALs are explained in detail, but DAALs are not; a more detailed explanation of DAALs is needed. Personally, I think a table summarizing MAALs would be a better option.
Response: disomic alien addition lines (DAALs). Table 2 is added for summarizing MAALs / DAALs.
Minor comments
Brassica juncea, B. napus, B. carinata (lines 84-85) must be italic. Same corrections are required in numerous other parts in the manuscript
Response: These are corrected.
Reviewer 2 Report (New Reviewer)
This Viewpoint article focuses mainly on the cytology of this crucifer species.
Throughout there are minor mistakes (polyploidy is used instead of polyploid eg title line 1; abstract line 16, section 3 title line 82; tetraploidy instead of tetraploid line 130. There is inconsistency in the use of italics with some words unexpectedly given in italics (lines 174-175) and many botanical names from line 84 onwards not given in italics.
Structurally this was difficult to read with a lot of text that could have been broken up by figures redrawn from the source literature. For example:
Section 1 (introduction) would have benefited from a map showing the distribution of the different species.
Section 2 (phylogenetics) is not helpful without a visual representation of the phylogenetic tree.
Section 3 is clearly where the authors have most expertise, and does include some chromosome images. Perhaps a schematic cartoon demonstrating the likely steps (duplication, translocation, chromosome loss etc) during the evolution of the Orychogramus and Brassica genomes from their ancestral diploid would be helpful.
Much of section 3, including figure 2, is actually describing hybrids and should be in section 4. The current layout with the image for Figure 2 separated from its legend made it difficult to follow the narrative.
Section 5 would be better if the attributes of MAAL1-9 and DAAL1-9 were tabulated; perhaps with some images included of the different leaf (MAAL1) and flower (MAAL4, DAAL4) forms. Mention of the OvPAP2 gene (line 231-232) felt artificial and completely out of context; perhaps this could be improved by rewording.
Section 6 is titled 'conclusion' but in fact is mainly about the unique biochemistry of the species, which is the main area of novelty and the rationale behind attempting to domesticate and breed new varieties.
There appeared to be a large number of self-citations, but this is partly a result of the species being mainly restricted to China.
Overall, this gave a rather unbalanced view of the potential for this new crop, skewed towards its chromosome pairing behaviour and cytology, without leaving the reader with much insight.
Author Response
This Viewpoint article focuses mainly on the cytology of this crucifer species.
Throughout there are minor mistakes (polyploidy is used instead of polyploid eg title line 1; abstract line 16, section 3 title line 82; tetraploidy instead of tetraploid line 130. There is inconsistency in the use of italics with some words unexpectedly given in italics (lines 174-175) and many botanical names from line 84 onwards not given in italics.
Response: polyploid is one noun, and polyploidy is one adjective. Some errors are corrected.
Structurally this was difficult to read with a lot of text that could have been broken up by figures redrawn from the source literature. For example:
Section 1 (introduction) would have benefited from a map showing the distribution of the different species.
Response:we think Table 1 is enough for the distributions of the these species, as the types with 2n=20, 22, 48 have the limited regions with no more descriptions later.
Section 2 (phylogenetics) is not helpful without a visual representation of the phylogenetic tree.
Response: Figure 1 is added for the phylogenetical relationship and karyotype evolution of Brassica and Orychophragmus.
Section 3 is clearly where the authors have most expertise, and does include some chromosome images. Perhaps a schematic cartoon demonstrating the likely steps (duplication, translocation, chromosome loss etc) during the evolution of the Orychogramus and Brassica genomes from their ancestral diploid would be helpful.
Response: Thanks for the good suggestion. Figure 1 is added for the phylogenetical relationship and karyotype evolution of Brassica and Orychophragmus.
Much of section 3, including figure 2, is actually describing hybrids and should be in section 4. The current layout with the image for Figure 2 separated from its legend made it difficult to follow the narrative.
Response: Thanks for the good suggestion. Original Figure 1 and 2 are combined, and some figures are removed.
Section 5 would be better if the attributes of MAAL1-9 and DAAL1-9 were tabulated; perhaps with some images included of the different leaf (MAAL1) and flower (MAAL4, DAAL4) forms. Mention of the OvPAP2 gene (line 231-232) felt artificial and completely out of context; perhaps this could be improved by rewording.
Response: Table 2 is added for summarizing MAALs / DAALs. The description of the OvPAP2 gene is reworded.
Section 6 is titled 'conclusion' but in fact is mainly about the unique biochemistry of the species, which is the main area of novelty and the rationale behind attempting to domesticate and breed new varieties.
Response: The Title is changed as:A Potential Oil Crop.
There appeared to be a large number of self-citations, but this is partly a result of the species being mainly restricted to China.
Response: The cytogenetic studies of this species are mainly from our group.
Overall, this gave a rather unbalanced view of the potential for this new crop, skewed towards its chromosome pairing behaviour and cytology, without leaving the reader with much insights
Response: Previous studies are mainly on the cytogenetics, and the novel fatty acids are recently found, which makes the species as one new oil crop.
Reviewer 3 Report (New Reviewer)
The work by the authors is an interesting commmunication about the characterization of the Orychophragmus violaceous genome. As pointed by the authors, this plant species is becoming very attractive for researchers interested in plant seed oils and biofuels because its high amount of oil in the seed as well as its interesting fatty acid composition, enriched in very long chain fatty acids, which are good for high-quality biofuels. The characterization of the O. violaceous genome through the analysis of hybrids with other brassicaceae like oleracea or rapa is very interesting and provides information of groups or clusters of genes that could be responsible of these interesting plant seed-oil traits. In this sense, I miss some additional information that would improve the quality of the paper, like a fatty acid composition analysis and seed oil content data of the hybrids in comparison with the WT O. violaceous or the Brassicaceae used for hybridization. In parallel, although the main justification for the interest of working in O. violaceous is its seed oil content and composition, no genetic information of genes involved in seed-oil biosynthesis (i.e. genes duplicated in the hybrids, genes retained, genes that dissapeared, homology analysis of O. violaceous genes with respect to those from other brassicaceae...) is provided in the work. Apart the information of Figure 2, which is a general comparison, I would have expected a more detailed analysis of the genes involved in seed-oil lipid biosynthesis metabolism. The paper rests at the cytological level through some GISH analysis but further genomic information would improve the quality of the work.
Additionally, Figure 1 has very poor quality, it would be neccessary to improve it. Moreover, authors should revise the figure legend: is d) a DAPI stain?; is d) the same that a) which clearly shows a DAPI staining?, according to the legend, what is the difference between a) and d)?.
Author Response
Response: As the hybrids of O. violaceus with Brassica species are unstable cytologically, no progenies with alien introgressions and the changed fatty acid compositions are obtained, and then no genetic analyses on the seed-oil biosynthesis in the progenies. The seed-oil lipid biosynthesis metabolism in O. violaceus is elucidated for the novel dihydroxy fatty acids, based on the biochemistry and genome sequencing. The following sentences are added in the final paragraph:
Its Fatty Acid Desaturase2 (FAD2) and Fatty Acid Elongase1 (FAE1) were found to develop specific enzymatic activities critical for dihydroxy fatty acids biosynthesis [38]. The enzyme encoded by OvFAD2-2 functions as a fatty acid hydroxylase to generate the terminal hydroxyl group of two acids, instead of catalyzing fatty acid desaturation. Additionally, the functional variants encoded by OvFAE1-1 produced the carboxyl-terminal hydroxyl group of dihydroxy fatty acids through a “discontinuous elongation” process. Based on the genome and the transcriptome data of multistage seed development, it was predicted that OvDGAT1-1 and OvDGAT1-2 (DGAT, diacylglycerol acyltranferase) were the candidate genes regulating the storage of dihydroxy fatty acids in its seeds. From the results of genome sequencing, the formation of dihydroxy fatty acids was hypothesized to occur after the genome tetraploidization event [7].
Additionally, Figure 1 has very poor quality, it would be neccessary to improve it. Moreover, authors should revise the figure legend: is d) a DAPI stain?; is d) the same that a) which clearly shows a DAPI staining?, according to the legend, what is the difference between a) a
Response: Figure 1 and 2 are merged, and some pictures are deleted.
Round 2
Reviewer 2 Report (New Reviewer)
This is much improved . One remaining correction I would suggest would be to redraw Figure 3 so that the bottom connecting line is horizontal - a triangle with a flat base.
Author Response
This is much improved. One remaining correction I would suggest would be to redraw Figure 3 so that the bottom connecting line is horizontal - a triangle with a flat base.
Response: Yes, the figure is redrawn with a flat base.
Reviewer 3 Report (New Reviewer)
I have read the answer of the authors and i stll have some problems. authors claim that the oil and fatty analyais of the mutants was not interesting since the hybrids were not cytologically stable. I can understand that these hybrids could not be stable within generations but, I sill feel that the analysis could be done in the generations that are analyzed. Moreover, if they are not stable, are the genetic conclusions of the work meaningfull?.
On the other hand, I still feel that additional information should be needed. In their response to the review, the authors mention potential dgat activities, but no data are shown about this. dagt aqctivities cannot be mentioned in the terms used by tha ajuthors without any data. Finally, instead of answering to the errors or problemas in Figure 1, now a completely new figure 1 merging old figs 1 and 2 is shown. Why have they merged it?.
Author Response
I have read the answer of the authors and i stll have some problems. authors claim that the oil and fatty analyais of the mutants was not interesting since the hybrids were not cytologically stable. I can understand that these hybrids could not be stable within generations but, I sill feel that the analysis could be done in the generations that are analyzed. Moreover, if they are not stable, are the genetic conclusions of the work meaningfull?.
Response: We found some changes of the contents of the fatty acids in the Brassica parents-like progenies, but the kinds of the fatty acids had no changes. The progenies after several generations recovered the chromosome number and karyotype, and some fragment variations are detected.
On the other hand, I still feel that additional information should be needed. In their response to the review, the authors mention potential dgat activities, but no data are shown about this. dagt aqctivities cannot be mentioned in the terms used by tha ajuthors without any data. Finally, instead of answering to the errors or problemas in Figure 1, now a completely new figure 1 merging old figs 1 and 2 is shown. Why have they merged it?.
Response: The DGAT activities are cited from the Reference [7], which is derived from the genomic analysis and needs further study. The sentences are revised as: Based on the genome and the transcriptome data of multistage seed development, it was predicted that OvDGAT1-1 and OvDGAT1-2 (DGAT, diacylglycerol acyltranferase) were the candidate genes regulating the storage of dihydroxy fatty acids in its seeds, but further study is needed to confirm this.
The two figures are merged, to better present and compare the chromosome characteristics of O. violacues itself and in hybrids.
Round 3
Reviewer 3 Report (New Reviewer)
I understand and agree with the comments of the authors.
This manuscript is a resubmission of an earlier submission. The following is a list of the peer review reports and author responses from that submission.
Round 1
Reviewer 1 Report
The manuscript describes the genome structure and relationship of Orychophragmus violaceus. The topic in this study is interesting. However, in my opinion, I felt some of the authors' descriptions difficult to follow; I suspect a reader less familiar with the topic might have even greater difficulties. I would like to suggest some modifications as follows.
The genus Oryphophragmus of Brassicaceae described in the manuscript should be summarized in one Table, including chromosome numbers, to make it easier for readers to understand.
The relationship of hybrids should be summarized in a diagram. I believe it must help the reader to understand it as well.
Fig.1 and 2. The sub-numberings by A1, A2 seems unusual. The authors should follow the instruction.
Fig. 1A2. It appears to be a green signal, not a red signal from Cy3.
Reviewer 2 Report
In the manuscript entitled: "Orychophragmus violaceus Endemic to China: With Beautiful Flowers, Unique Oil and Large Genome" (Number: plants-2039341), authors stated “This review summarizes the research progress of the genome structure and evolution, cytogenetics of the intergeneric hybrids with Brassica crops, and the synthesis of the dihydroxy fatty acids in O. violaceus.” In my opinion, the title of this manuscript is inadequate for the data presented – how it is connected to „With Beautiful Flowers, Unique Oil”. Authors mixed genetic issue with oil content, two totally different aspects.
In general the manuscript has low scientific value, it is very descriptive and authors did not provide audience with profound knowledge about genetics of Orychophragmus violaceus. All data are provided in a very general way, and there is no critical summary; for example the phylogenetic analysis should be shown to provide readers with genetic relationship of O. violaceus with other plant species. Additionally, another example id section #6, which is ill-conceived. I do not understand purpose of this section entitled: ‘Aboundant Dihydroxy Fatty Acids‘ in connection to genetic background provided in the publication. It looks like random addition; yes, it can be subject, but on its own, not like short addition to genetic data. Finally, be more specific and science oriented, what does it means ‘With Beautiful Flowers’ - beauty it is very subjective issue, and should be taken with great care especially with genetic background.
In conclusion, in my opinion, the scientific value of the presented manuscript is very low.